# The Geriatric Nutritional Risk Index Predicts Prognosis in Japanese Patients with LATITUDE High-Risk Metastatic Hormone-Sensitive Prostate Cancer: A Multi-Center Study

**DOI:** 10.3390/cancers15225333

**Published:** 2023-11-08

**Authors:** Taku Naiki, Kiyoshi Takahara, Hiromitsu Watanabe, Keita Nakane, Yosuke Sugiyama, Takuya Koie, Ryoichi Shiroki, Hideaki Miyake, Takahiro Yasui

**Affiliations:** 1Department of Nephro-Urology, Graduate School of Medical Sciences, Nagoya City University, Nagoya 467-8601, Japan; phsugi@sunprom.med.nagoya-cu.ac.jp (Y.S.); yasui@med.nagoya-cu.ac.jp (T.Y.); 2Department of Urology, Fujita Medical University, Nagoya 470-1192, Japan; takahara@fujita-hu.ac.jp (K.T.); rshiroki@fujita-hu.ac.jp (R.S.); 3Department of Urology, Hamamatsu University School of Medicine, Hamamatsu 431-3125, Japan; urohiro@hama-med.ac.jp (H.W.); hmiyake@hama-med.ac.jp (H.M.); 4Department of Urology, Gifu University, Gifu 501-1112, Japan; nakane.keita.k2@f.gifu-u.ac.jp (K.N.); koie.takuya.h2@f.gifu-u.ac.jp (T.K.)

**Keywords:** abiraterone acetate, aged, androgen antagonists, bicalutamide, nutrition assessment

## Abstract

**Simple Summary:**

The geriatric nutritional risk index (GNRI) is used as a prognostic factor in a variety of cancers. We aimed to evaluate the prognostic significance of the pretreatment GNRI and retrospectively compared androgen deprivation therapy (ADT) plus up-front abiraterone acetate (AA) or bicalutamide in patients with metastatic hormone-sensitive prostate cancer (mHSPC) using large multi-institutional data. We found that ADT plus abiraterone may have advantages over ADT plus bicalutamide. In addition, our analysis revealed the importance of prolonged time to castration-resistant prostate cancer even in the era of upfront androgen receptor axis target therapy or docetaxel. It also highlighted the prognostic significance of the pretreatment GNRI for patients with LATITUDE high-risk mHSPC treated with upfront AA plus ADT.

**Abstract:**

Malnutrition is associated with prognosis in cancer. The geriatric nutritional risk index (GNRI), based on the ratio of actual to ideal body weight and also serum albumin level, is a simple screening tool for assessing nutrition. We investigated the GNRI as a prognostic factor for oncological outcomes in patients with high-risk metastatic hormone-sensitive prostate cancer (mHSPC) using a Japanese multicenter cohort. This study included a total of 175 patients with LATITUDE high-risk mHSPC, of whom 102 had received androgen deprivation therapy (ADT) plus upfront abiraterone acetate, and 73 had received ADT plus bicalutamide (Bica), from 14 institutions associated with the Tokai Urologic Oncology Research Seminar. Patients were classified into GNRI-low (<98) or GNRI-high (≥98) groups. The GNRI was based on the body mass index and serum albumin level. Kaplan–Meier analysis revealed that the median overall survival (OS) of a GNRI-low group (median 33.7 months; 95% confidence interval [CI]: 26.2–not reached [NR]) was significantly worse than that of a GNRI-high group (median: NR; 95% CI: NR–NR; *p* < 0.001). Multivariate analysis identified Bica and low GNRI (<98) as independent prognostic factors for reduced times to both castration-resistant prostate cancer and OS, and, therefore, a poor prognosis. Our findings indicate the GNRI may be a practical prognostic indicator in the evaluation of survival outcomes in patients with LATITUDE high-risk mHSPC.

## 1. Introduction

Androgen deprivation therapy (ADT) is given to patients with metastatic prostate cancer (PCA). However, a good initial response to ADT is often followed by the development of progressive castration-resistant prostate cancer (CRPC), leading to death in most patients with PCA [1]. In metastatic hormone-sensitive prostate cancer (mHSPC), the survival benefits of combined ADT and second-generation androgen receptor axis-targeted agents (ARATA), which include abiraterone acetate (AA) [2,3] or docetaxel [4], when compared to ADT alone were highlighted in several recent Phase III trials. For high-risk patients with PCA, including LATITUDE high-risk individuals, recent evidence has led to a treatment framework of intense therapy given at an earlier treatment phase. Consequently, the prognosis of patients with mHSPC has gradually improved [5]. The LATITUDE study revealed that in patients with a large tumor volume, ADT together with AA led to improved radiologic progression-free survival (PFS) and overall survival (OS) compared to ADT alone [6]. Thus, various clinical guidelines strongly recommend ADT treatment combined with upfront AA [7].

For patients with mHSPC, combined androgen blockade (CAB) is one of several initial treatments given, especially in East Asian countries like Japan [8]. Although conventional CAB using bicalutamide (Bica) remains the prevailing choice for Japanese physicians, a direct prospective comparison of upfront AA plus ADT and CAB, particularly focused on data from Asian patients, might reveal new strategies for disease treatment. In Japanese patients, we and others have directly compared upfront AA plus ADT and CAB in those with high-risk mHSPC [9,10,11,12,13,14]. In terms of PFS, upfront AA plus ADT was deemed superior to CAB in all prior studies; however, both our group [9] and another group [10] found that the two treatments showed no significant difference in OS. In addition, whether all LATITUDE patients with high-risk mHSPC should be treated with upfront AA plus ADT remains unclear. Thus, the best treatment choice for a specific patient remains unresolved.

Malignancies in patients are often accompanied by malnutrition such that medical nutritional therapy has become a necessary part of multidisciplinary anticancer treatment programs [15]. Various nutritional assessment tools can identify survival-related prognostic indicators in multiple malignancies, including PCA [16]. The geriatric nutritional risk index (GNRI) is an excellent tool that is used to assess nutrition and is calculated using the ratio of actual to ideal body weight in addition to the albumin (Alb) level; it is one such potential prognostic indicator [17]. We recently described the utility of GNRI as a prognostic indicator that predicted survival outcomes in patients with bladder cancer treated with immunochemotherapy [18].

However, to date, no reports exist on the relationship between GNRI and outcomes for patients with LATITUDE high-risk mHSPC treated with upfront AA plus ADT or CAB. Therefore, we evaluated whether GNRI could be used as a prognostic indicator in such patients.

## 2. Materials and Methods

### 2.1. Patients

For this investigation, we collected further data from a previously studied cohort [8] of a total of 175 patients with mHSPC who were designated as “high risk” in the LATITUDE trial. Between January 2018 and September 2020, patient data were accrued at our institution as well as affiliated hospitals including Nagoya City University Graduate School of Medical Sciences, Hamamatsu University School of Medicine, Fujita Health University School of Medicine, and Gifu University associated with the Tokai Urologic Oncology Research Seminar group. Inclusion criteria for patients were as follows: those who underwent treatment with ADT together with AA taken orally (102 patients; 1000 mg once a day) + prednisolone (upfront AA plus ADT treatment), or ADT and Bica taken orally (73 patients; 80 mg once a day), also known as CAB treatment. Criteria from Prostate Cancer Clinical Trials Working Group 3 were used to define biochemical, radiographic or clinical progressive disease.

### 2.2. Oncological Assessment

Patients were treated until radiographic or clinical disease progression was noted as well as an increased prostate-specific antigen (PSA) level. Overall survival was determined from the time period between the start of first therapy and death due to all-cause mortality. A diagnosis of CRPC was based on radiographic or PSA progression, and time to CRPC (TTCR) was measured from the start of first treatment until the time of a CRPC diagnosis. The start of initial treatment until a second/subsequent tumor progression in next-line treatment was defined as the time to second progression (PFS2).

### 2.3. Data Collection

The medical records of patients of the abovementioned institutions were used to extract patient information regarding age, height, weight, serum blood variables, ECOG–PS, initial PSA levels, the Gleason score from a prostate biopsy, and PSA kinetics. In addition, blood variables at treatment initiation, including alkaline phosphatase, C-reactive protein (CRP), Alb, and lymphocyte and neutrophil counts, were also collected. As previously reported [19], patients were grouped into three subgroups based on TTCR as outlined: 0–12, 12.1–18, and ≥18.1 months.

### 2.4. Nutritional Assessment by GNRI

Patients were classified into GNRI-low (<98) or GNRI-high (≥98) groups. The formula used for GNRI values was: 1.489 × serum Alb level (g/L) + 41.7 × (actual body weight [kg]/ideal body weight [kg]). If actual body weight was greater than ideal body weight the ratio of these two factors was set to one. The formula for ideal body weight is: ideal body weight = 22 × height (m^2^) [9,20].

### 2.5. Ethics Approval

This study was undertaken with the approval of the ethics committees of all universities belonging to this group (approval nos. 60-21-0018, 2021-042, 21-051, HM20-465) and websites outlined opt-out information for patients. The investigation was conducted according to the Declaration of Helsinki (2013).

### 2.6. Statistics

To evaluate differences in categorical parameters, we used Fisher’s exact test or a Mann–Whitney *U* test, when appropriate, as an ordinal scale. After receiver operating characteristic (ROC) curve analysis of the total cohort with regard to CRPC development, new cutoff values for parameters were determined as drawn using a Youden Index. Cumulative rates of survival were estimated from Kaplan–Meier curves. Log-rank tests were used to determine significant differences between curves. Univariate and multivariate analyses were based on Cox proportional hazard regression analyses. Variables that were clinically important factors were used to predict TTCR and OS. Data were evaluated with the use of EZR software (Saitama Medical Center, Jichi Medical University, Yakushiji, Japan).

## 3. Results

### 3.1. Characteristics and Outcomes of Patients by Further Follow-Up Study

A total of 175 patients, traceable from a previous analysis [8], were enrolled in this retrospective study. Of these, 73 and 102 patients received either CAB treatment or upfront AA plus ADT treatment, respectively. The two groups showed a significantly different median follow-up period of between 33.7 months for the CAB group vs. 26.2 months for the upfront AA plus ADT group (*p* < 0.001). As shown in Appendix A, peripheral blood markers and clinical parameters were not statistically different between the two patient groups. Furthermore, the median time to PSA nadir and rate of PSA decline >50% were similar. However, CRPC was found to occur more among patients in the CAB group when compared to those in the upfront AA plus ADT group (47/73; 64.4% for CAB vs. 31/102; 30.4% for upfront AA plus ADT, *p* < 0.001). Upfront AA plus ADT treatment led to a significantly prolonged median TTCR compared to CAB treatment (not reached [NR]; 95% confidence interval [CI]: NR–NR vs. 12.9 months; 95% CI: 9.2–22.0, *p* < 0.0001; Appendix A). In addition, PFS2 in the upfront AA plus ADT group was significantly superior as shown in Appendix A (median NR, 95% CI: NR–NR) compared to the PFS2 of the CAB group (median: NR, 95% CI: 25.7–NR; Appendix A; *p* < 0.01). However, OS in the CAB group (median NR, 95% CI: 32.9–NR) did not significantly differ from that in the upfront AA plus ADT group (median NR, 95% CI: NR–NR; Appendix A).

Thus, upfront AA plus ADT treatment led to decreased CRPC, and prolonged median TTCR and PFS2 compared to CAB treatment, although OS did not differ.

### 3.2. Setting New Cutoff Values and Prognostic Analysis Focusing on Serum Biomarkers

For the setting of new cutoff values for prognostic factors, clinical factors, including age, initial PSA, Alb, GNRI, CRP, and the neutrophil-to-lymphocyte ratio (NLR), were analyzed using ROC curves. New cutoff values were determined to be: 76 years of age, an initial PSA level of 82.4 ng/mL, an Alb level of 3.6 g/L, a GNRI of 98, a CRP level of 1.22 mg/dL, and an NLR of 2.003 as shown in Figure 1. Of the 175 patients, 66 were grouped within the GNRI-low group and 109 within the GNRI-high group. Baseline clinical oncological parameters were similar in the two groups (Table 1). However, the median body mass index (BMI), proportion of patients with better ECOG-PS, as well as serum Alb levels were significantly greater in the GNRI-high compared with GNRI-low group. Median age and median initial PSA levels were found to be significantly greater in the GNRI-low compared to GNRI-high group. Consistent with these data, CRPC development was found to be significantly lower in the GNRI-high compared to GNRI-low group (42/109 [38.5%] vs. 36/66 [54.5%] in GNRI-high vs. -low groups, respectively). Additionally, analysis of the whole cohort revealed that segregating patients based on age, initial PSA, and NLR status did not reveal significant differences in OS (Figure 2a,b,d). However, the median OS of the GNRI-low group (median 33.7 months; 95% CI: 26.2–NR) was significantly worse than that of the GNRI-high group (median NR; 95% CI: NR–NR; Figure 2f; *p* < 0.001). Concerning CRP, a significant difference in OS when comparing CRP-high and -low groups was also evident (Figure 2e). Furthermore, also in the analysis of cohorts receiving upfront AA plus ADT treatment, patients with high GNRI showed significantly prolonged OS compared to those with a low GNRI (*p* < 0.001; Figure 2h), similar to CRP (*p* < 0.05; Figure 2g).

In summary, of the total cohort, a greater proportion of patients in the GNRI-high group showed a significantly greater BMI and serum Alb level, better ECOG-PS, lower median age, and initial PSA level, consistent with lower CRPC development and longer survival rates for this group. Patients undergoing upfront AA plus ADT treatment also showed significantly longer survival rates if they had a high GNRI.

### 3.3. Identification of Prognostic Factors for TTCR and OS

Concerning Alb, OS was statistically significant when comparing Alb-high and -low groups (*p* < 0.001; Figure 2c). The GNRI was calculated using a serum Alb-based formula when performing univariate and multivariate analyses. These two items strongly correlated and, therefore, both were not simultaneously included. When compared with GNRI, most patients that were divided by an Alb level cutoff were categorized as being in the Alb-high group (134 and 41 in high and low groups, respectively, for Alb; vs. 109 and 66 in high and low groups, respectively, for GNRI). Considering the substantial contribution of the Alb level to predicting OS in this study, and recent evidence on the efficacy of complex biomarkers, including serum Alb [21,22], the GNRI was therefore selected for this study. With regard to prolonged TTCR, the following were revealed in univariate analysis to be independent prognostic factors when initial treatment was started: existence of symptoms (HR: 1.87, 95% CI: 1.16–3.03), upfront AA treatment (HR: 0.38, 95% CI: 0.24–0.60), low CRP level (HR: 0.45, 95% CI: 0.28–0.74), and high GNRI (HR: 0.44, 95% CI: 0.28–0.69). Additionally, for prolonged TTCR, multivariate analysis identified significant prognostic factors: upfront AA treatment (HR: 0.39, 95% CI: 0.23–0.66) and high GNRI (HR: 0.38, 95% CI: 0.21–0.67) (Table 2). Furthermore, high GNRI was also identified as the sole prognostic indicator of OS both in univariate and multivariate tests (HR: 0.35, 95% CI: 0.19–0.64, in univariate, and HR: 0.45, 95% CI: 0.21–0.96, in multivariate, respectively; Table 3).

Thus, the univariate analysis identified symptoms, upfront AA treatment, low CRP, and high GNRI to be significantly associated with worse survival rates. Multivariate analysis identified the variables of upfront AA treatment and high GNRI as being significantly associated with TTCR; high GNRI was also significantly associated with OS.

### 3.4. Analysis of OS Based on TTCR

As reported above, elongation of the follow-up period did not reveal a statistically significant difference in OS between patients receiving CAB and those receiving upfront AA plus ADT treatment. Therefore, an evaluation of OS based on TTCR classification was made. An analysis of the total cohort revealed that the median OS from diagnosis was 24.7 (95% CI: 16.6–31.9), NR (95% CI: 34.2–NR), and NR (95% CI: NR–NR) months in those patients with a TTCR of 0–12, 12.1–18, and ≥18.1 months, respectively (Figure 3a). Differences in OS between the three groups were statistically significant. In addition, it was observed that OS was proportional to TTCR in the upfront AA plus ADT treatment group (Figure 3b). Figure 3c shows the results of our analysis of “upfront AA plus ADT or not” and “GNRI high or low” as prognostic factors of PFS. The PFS showed significant differences between the three groups classified according to the positive number of these two independent factors (median 7.5 months, 95% CI: 4.1–NR in GNRI-low patients treated with CAB; median 14.9 months, 95% CI: 12.2–NR in GNRI-low patients treated with upfront AA plus ADT, or in GNRI-high patients treated with CAB; median NR months, 95% CI: NR–NR in GNRI-high patients treated with upfront AA plus ADT).

Thus, a shorter TTCR in the total cohort and upfront AA plus ADT treatment group led to reduced OS. When patients with various TTCR rates were further segregated, PFS was found to be prolonged for those patients with a high GNRI and/or treated with upfront AA plus ADT.

## 4. Discussion

Malignancies in patients are often accompanied by patients experiencing malnutrition [15]. The GNRI is used to assess nutrition and is a potential prognostic indicator of the risk of morbidity and mortality [17]. However, to date, no reports exist on the relationship between GNRI and outcomes for patients with LATITUDE high-risk mHSPC treated with upfront AA plus ADT or CAB.

In this study on Japanese patients with LATITUDE high-risk mHSPC, although OS was unchanged over a longer follow-up period, we found PFS2 after upfront AA plus ADT treatment was significantly superior to that of CAB treatment. In the current era of upfront treatment given to patients showing high-risk mHSPC, the clinical importance of a prolonged TTCR has not been conclusively established. In this analysis, differences in OS between three groups of patients, distinguished according to TTCR values, were statistically significant. In addition, OS was directly proportional to TTCR in the upfront AA plus ADT treatment group. To our knowledge, only one study [23] has described worse OS in patients with a TTCR < 12 months, with OS gradually increasing as the TTCR period increased. However, this study was performed on a heterogenous population and a variety of agents was used. Previously, before the era of upfront treatment, we grouped patients with mHSPC into four groups based on the length of TTCR in order to compare clinicopathological characteristics. We found that shorter TTCR in patients was associated with an unfavorable OS [19]. Here, we show a longer TTCR favored improved OS both in total and upfront AA with ADT cohorts. We, therefore, conclude that even in the era of intensified upfront therapy, TTCR should be extended for the maximum time possible so as to reach the best prognostic outcomes in patients with LATITUDE high-risk mHSPC.

Even in the era of intensified upfront treatment for mHSPC, it is unclear whether all patients with LATITUDE high-risk mHSPC should be treated upfront with ARATA. Furthermore, it is important to identify patients most likely to benefit from upfront AA plus ADT or CAB using a suitable indicator superior to other variables, such as Gleason score, clinical stage, or ECOG–PS [9]. Recently, several molecular mechanisms to explain the role of inflammation in PCA have been proposed. These include cellular turnover, induction of genomic and cellular environment that leads to replication, and activation of tissue repair [24]. The NLR may be a reliable serum biomarker of inflammation since elevated NLR at initial treatment is predictive of poor OS rates in patients with mHSPC [25]. Additionally, before the era of intensified therapy for mHSPC, PSA and CRP independently predicted poorer cancer-specific survival in a patient cohort with mHSPC receiving ADT alone [26]. Although several studies recently linked CRP levels to CRPC [27], recent data linking CRP to prognosis in castration-sensitive PCA are lacking.

In our current study, we evaluated three immune-nutritional parameters as potential prognostic factors. Of these, high GNRI was superior to low GNRI as a prognostic indicator of both PFS and OS; this was the case for both the total cohort and the upfront AA plus ADT treatment cohort. Furthermore, after univariate and multivariate analyses, high GNRI was identified as the sole prognostic indicator that predicted both PFS and OS. Additionally, in the analysis of PFS focusing on two prognostic factors (upfront AA plus ADT or not, and GNRI high or low), significant differences were noted in PFS among the three TTCR groups classified according to the positive number of these two independent factors.

The GNRI is thought to reflect mortality in elderly patients as well as those on hemodialysis and with cardiovascular disease. This notion has also been used for patients with various cancers, including lung, gastrointestinal, and urothelial cancers [18,28,29,30]. Markers of nutritional status, including a decreased serum Alb level or BMI, have been associated with a poor oncological outcome in metastatic prostate cancer [31]. However, only one report exists of an association between high levels of GNRI and a better prognosis in mHSPC [32]. As a novelty, our study suggested the superiority of GNRI compared to NLR or CRP as a biomarker in patients with LATITUDE high risk, and also suggested that the GNRI could be used to aid in patient selection in the upfront treatment of such patients. In addition, our data also suggest that all patients with LATITUDE high-risk and high GNRI levels should receive upfront AA with ADT instead of CAB as initial treatment. Furthermore, considering the poor prognosis, upfront AA with ADT should be strongly recommended for patients with low GNRI. Further prospective studies on upfront intensified treatment are necessary to predict subsequent responses and survival outcomes.

Our study had several limitations. Although the data originated from among the largest cohorts of patients with LATITUDE high-risk mHSPC to receive upfront AA plus ADT or CAB, it is a retrospective analysis with the usual shortcomings of selection bias and small sample sizes. Second, both PFS and PFS2 were significantly superior in the upfront AA plus ADT cohort compared to the CAB cohort. However, though the follow-up period was extended by about 14 months from a previous study, a difference in OS was not observed between these two groups, which may be due to a clear difference in the median follow-up period. Third, the GNRI is a formula that incorporates the Alb level and is, therefore, not convenient compared with Alb alone. In this analysis, the patient population divided by Alb alone was imbalanced as described above; GNRI was selected as a biomarker. However, evaluating the superiority of GNRI compared with Alb was not an aim of this study. Finally, we could not perform second-line treatment-specific TTCR analyses because of limited sample sizes and baseline characteristics that differed between second-line treatment groups. A long-term follow-up study is required in the future to support our study conclusions.

## 5. Conclusions

In summary, we demonstrated that the GNRI may be a practical prognostic indicator of survival outcomes in patients with LATITUDE high-risk mHSPC. In addition, even in the era of upfront intensified ARATA or docetaxel, prolonging TTCR is required to achieve the best prognostic outcomes in patients with LATITUDE high-risk mHSPC. These data provide information that can aid in the selection of the first therapies for mHSPC patients, including those in Japan.

## Figures and Tables

**Figure 1 cancers-15-05333-f001:**
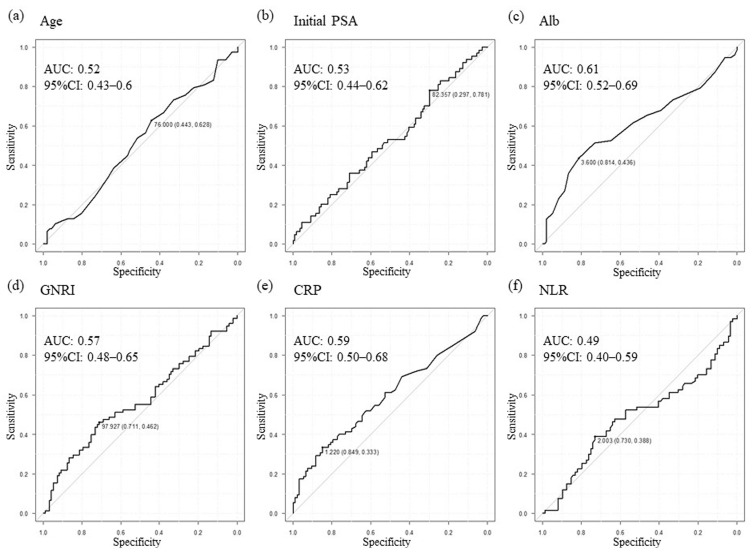
Cutoff value setting using ROC curves analysis of indicators for CRPC development as outcome variables in the total cohort. Age (**a**), Initial PSA (**b**), Alb (**c**), GNRI (**d**), CRP (**e**), NLR (**f**). Alb, Albumin; CRP, C-reactive protein; CRPC, Castration-resistant prostate cancer; GNRI, Geriatric nutritional risk index; NLR, Neutrophil-to-lymphocyte ratio; PSA, Prostate-specific antigen; ROC, Receiver operating characteristic.

**Figure 2 cancers-15-05333-f002:**
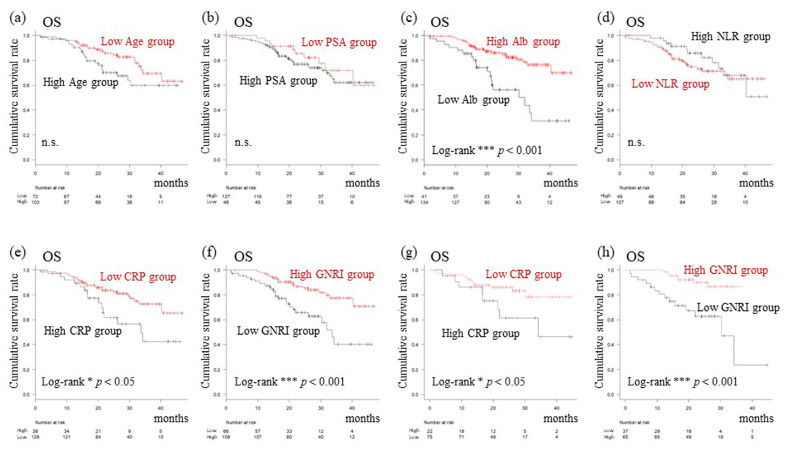
OS of Kaplan–Meier curves analyzing indicators in the total cohort (**a**–**f**) and patients treated upfront with AA plus ADT (**g**,**h**). Patients were categorized into low (<76, *n* = 72) or high (≥76, *n* = 103) age groups. OS (**a**). Patients were categorized into low (<82.4, *n* = 48) or high (≥82.4, *n* = 127) initial PSA groups. OS (**b**). Patients were categorized into low (<3.6, *n* = 41) or high (≥3.6, *n* = 134) Alb groups. OS (**c**). Patients were categorized into low (<2.003, *n* = 107) or high (≥2.003, *n* = 49) NLR groups. OS (**d**). Patients were categorized into low (<1.22, *n* = 128) or high (≥1.22, *n* = 38) CRP groups. OS (**e**). Patients were categorized into low (<98, *n* = 66) or high (≥98, *n* = 109) GNRI groups. OS (**f**). Patients were categorized into low (<1.22, *n* = 75) or high (≥1.22, *n* = 22) CRP groups. OS (**g**). Patients were categorized into low (<98, *n* = 37) or high (≥98, *n* = 65) GNRI groups. OS (**h**). Alb, Albumin; CRP, C-reactive protein; GNRI, Geriatric nutritional risk index; NLR, Neutrophil-to-lymphocyte ratio; OS, Overall survival; PFS, Progression-free survival; PSA, Prostate-specific antigen. * *p* < 0.05, *** *p* < 0.001. n.s., not significant.

**Figure 3 cancers-15-05333-f003:**
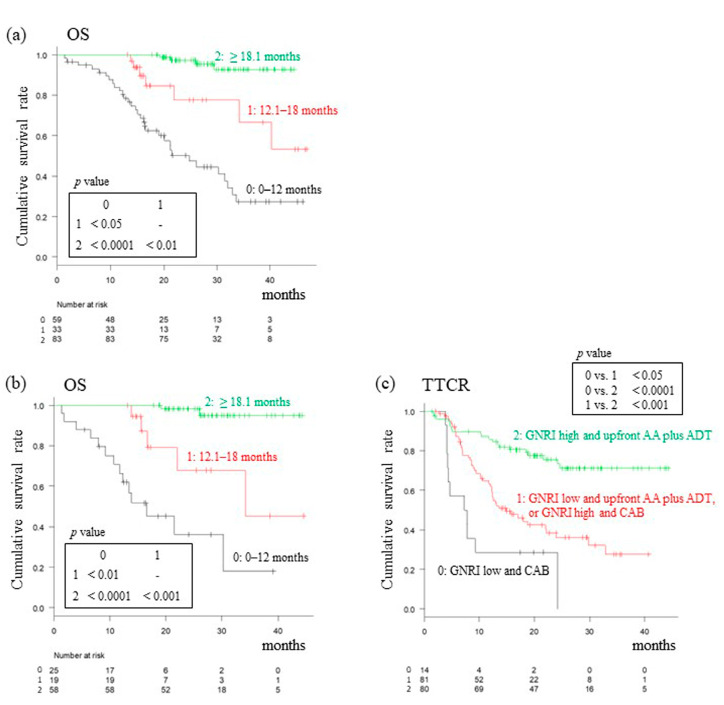
Kaplan–Meier curves for OS in LATITUDE high-risk mHSPC patients. OS from a diagnosis in LATITUDE high-risk mHSPC patients after initial treatment according to the TTCR in the total cohort (**a**) and upfront AA plus ADT group (**b**). “Upfront AA plus ADT or CAB” and “GNRI high or low” as prognostic factors for PFS (**c**). AA, Abiraterone acetate; ADT, Androgen deprivation therapy; CAB, Combined androgen blockade; CRPC, castration-resistant prostate cancer; mHSPC, Metastatic hormone-sensitive prostate cancer; OS, Overall survival; TTCR, Time to CRPC. *p* < 0.05, *p* < 0.01, *p* < 0.001, *p* < 0.0001, statistically significant n.s.: not significant.

**Table 1 cancers-15-05333-t001:** Characteristics and oncological outcomes of patients segregated by low and high GNRI status. * *p* < 0.05, ** *p* < 0.01, *** *p* < 0.001, statistically significant.

Characteristics		Low GNRI Group (*n* = 66)	High GNRI Group (*n* = 109)	*p* Value
Median age (range), years		77 (55–91)	72 (48–89)	<0.05 *
Median initial serum PSA levels (range), ng/mL		465.7 (4.7–16720)	189.6 (0.76–13433)	<0.01 **
Gleason score at initial diagnosis, *n* (%)	4 + 3	1 (1.5)	1 (0.9)	0.05
	4 + 4	26 (39.4)	41 (37.6)	
	4 + 5	15 (22.7)	42 (38.5)	
	5 + 4	13 (19.7)	19 (17.4)	
	5 + 5	11 (16.7)	6 (5.5)	
Clinical T stage	2a	0 (0)	4 (3.7)	0.35
	2b	3 (4.5)	5 (4.6)	
	2c	8 (12.1)	8 (7.3)	
	3a	12 (18.2)	28 (25.7)	
	3b	19 (28.8)	35 (32.1)	
	4	24 (36.4)	29 (26.6)	
Clinical N stage	N0	32 (48.5)	55 (50.5)	0.88
	N1	34 (51.5)	54 (49.5)	
Visceral metastasis, *n* (%)	no	45 (68.2)	72 (66.1)	0.87
	yes	21 (31.8)	37 (33.9)	
ECOG–PS, *n* (%)	0	36 (54.5)	91 (83.5)	<0.001 ***
	1	20 (30.3)	10 (9.2)	
	2	7 (10.6)	8 (7.3)	
	3	3 (4.5)	0 (0)	
Existence of symptoms, *n* (%)		44 (66.7)	58 (53.2)	0.09
Median BMI (range), kg/m^2^		20.4 (14.2–25.6)	23.7 (17.3–34.1)	NA
The use of bone modifying agents, *n* (%)		43 (65.2)	59 (54.1)	0.16
Median serum albumin level (range), g/dL		3.5 (1.5–4.3)	4.2 (2.8–6.6)	NA
Median serum ALP level (range), U/L		626 (136–11620)	333 (81–8300)	<0.01 **
Agents, combined with ADT	Bicalutamide	29 (43.9)	44 (40.4)	0.75
	Abiraterone	37 (56.1)	65 (59.6)	
PSA decline > 50%, *n* (%)		56 (84.8)	93 (85.3)	1
Median period to PSA nadir (range), days		143 (27–770)	330 (21–1215)	<0.001 ***
CRPC development, *n* (%)		36 (54.5)	42 (38.5)	<0.05 *

**Table 2 cancers-15-05333-t002:** Univariate and multivariate analyses were undertaken of baseline parameters and time to castration-resistant prostate cancer in all 175 patients treated with upfront AA plus ADT or CAB. * *p* < 0.05, ** *p* < 0.01, *** *p* < 0.001, **** *p* < 0.0001, statistically significant.

Parameters	Univariate			Multivariate		
	HR	95% CI	*p* Value	HR	95% CI	*p* Value
Ages, <76 vs. ≥76 years	1.14	0.72–1.80	0.58	1.36	0.79–2.32	0.27
Initial PSA levels, <82.4 vs. ≥82.4 ng/mL	0.62	0.36–1.06	0.08	0.92	0.49–1.72	0.80-
ECOG–PS, 1, 2, 3 vs. 0	1.53	0.94–2.48	0.08	1.06	0.59–1.91	0.84
Existence of liver metastasis, yes vs. no	1.66	0.41–6.78	0.48	3.18	0.72–14.1	0.13
Existence of symptoms, yes vs. no	1.87	1.16–3.03	< 0.05 *	1.61	0.90–2.88	0.11
Agent at the start of treatment, AA vs. Bica	0.38	0.24–0.60	<0.0001 ****	0.39	0.23–0.66	<0.001 ***
NLR, >2.003 vs. ≤2.003	0.79	0.48–1.30	0.36	0.46	0.25–0.84	<0.05 *
Serum CRP levels, <1.22 vs. ≥1.22 mg/dL	0.45	0.28–0.74	<0.01 **	0.65	0.37–1.12	0.12
GNRI, ≥98 vs. <98	0.44	0.28–0.69	<0.001 ***	0.38	0.21–0.67	<0.001 ***

**Table 3 cancers-15-05333-t003:** Univariate and multivariate analyses were undertaken of baseline parameters and overall survival in all 175 patients treated with upfront AA plus ADT or CAB. * *p* < 0.05, *** *p* < 0.001, statistically significant.

Parameters	Univariate			Multivariate		
	HR	95% CI	*p* Value	HR	95% CI	*p* Value
Ages, <76 vs. ≥76 years	0.57	0.31–1.04	0.07	0.67	0.34–1.30	0.23
Initial PSA levels, <82.4 vs. ≥82.4 ng/mL	0.74	0.36–1.50	0.40	1.47	0.65–3.31	0.35
ECOG–PS, 1, 2, 3 vs. 0	2.21	1.20–4.05	<0.05 *	1.82	0.90–3.71	0.10
Existence of liver metastasis, yes vs. no	1.56	0.21–11.4	0.66	1.27	0.15–10.50	0.83
Existence of symptoms, yes vs. no	1.47	0.78–2.79	0.23	1.07	0.52–2.22	0.86
Agent at the start of treatment, AA vs. Bica	0.83	0.45–1.51	0.53	1.07	0.54–2.10	0.85
NLR, >2.003 vs. ≤2.003	1.31	0.65–2.63	0.45	0.72	0.32–1.62	0.42
Serum CRP levels, <1.22 vs. ≥1.22, mg/dL	0.46	0.24–0.86	<0.05 *	0.53	0.23–1.22	0.14
GNRI, ≥98 vs. <98	0.35	0.19–0.64	<0.001 ***	0.45	0.21–0.96	<0.05 *

## Data Availability

Materials and raw data may be given after a request made to the corresponding author.

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
