# Peer review of "The Geriatric Nutritional Risk Index Predicts Prognosis in Japanese Patients with LATITUDE High-Risk Metastatic Hormone-Sensitive Prostate Cancer: A Multi-Center Study"

_cancers, 2023, doi:10.3390/cancers15225333_

Round 1

Reviewer 1 Report (Previous Reviewer 3)

Comments and Suggestions for Authors

In this study Authors aimed to test the Geriatric Nutritional Risk Index as an independent predictor of oncological outcomes in a Japanese cohort of patients with high-risk metastatic hormone-sensitive prostate cancer according to LATITUDE criteria receiving either androgen deprivation therapy and abiraterone acetate or androgen deprivation therapy plus bicalutamide. 

I see a substantial improvement in the Introduction and in the way to describe methods compared to the previous version.

However, again the manuscript contains tons of analyses and results presented with reference to so many figures that are not easy to read and interpret. Figures are often of low quality. In Figure 2, plot 2e and 2g seems to evaluate the same variable (CRP). My advice is to shortening the Results paragraph and maintaining an objective way to present results avoiding to discuss them here.

How did you perform multivariable analysis? Did you include in the model only variables that showed a statistical significance in the univariable analysis? This is an obsolete way to proceed. Maybe it is better to incorporate in a multivariable model what do you think could have a clinical effect on your outcome of interest. 

Author Response

Reviewer 1

In this study Authors aimed to test the Geriatric Nutritional Risk Index as an independent predictor of oncological outcomes in a Japanese cohort of patients with high-risk metastatic hormone-sensitive prostate cancer according to LATITUDE criteria receiving either androgen deprivation therapy and abiraterone acetate or androgen deprivation therapy plus bicalutamide. 

I see a substantial improvement in the Introduction and in the way to describe methods compared to the previous version.

However, again the manuscript contains tons of analyses and results presented with reference to so many figures that are not easy to read and interpret. Figures are often of low quality. In Figure 2, plot 2e and 2g seems to evaluate the same variable (CRP). My advice is to shortening the Results paragraph and maintaining an objective way to present results avoiding to discuss them here.

Thank you for the time spent in reviewing our article, and supplied valuable suggestion. In response to the reviewer’s comments, we changed all the figures to very high levels in resolution, and modified the description in Results for correct understanding in reader as below:

“With regard to prolonged TTCR, the following were revealed in univariate analysis to be independent prognostic factors when initial treatment was started: existence of symptoms (HR: 1.87, 95% CI: 1.16–3.03), upfront AA treatment (HR: 0.38, 95% CI: 0.24–0.60), low CRP level (HR: 0.45, 95% CI: 0.28–0.74), and high GNRI (HR: 0.44, 95% CI: 0.28–0.69). Addi-tionally, for prolonged TTCR, multivariate analysis identified significant prognostic fac-tors: upfront AA treatment (HR: 0.39, 95% CI: 0.23–0.66) and high GNRI (HR: 0.38, 95% CI: 0.21–0.67) (Table 2). Furthermore, high GNRI was also identified as the sole prognostic indicator of OS by both in univariate and multivariate tests ((HR: 0.35, 95% CI: 0.19–0.64, in univariate, and HR: 0.45, 95% CI: 0.21–0.96, in multivariate, respectively; Table 3).” In Results.

How did you perform multivariable analysis? Did you include in the model only variables that showed a statistical significance in the univariable analysis? This is an obsolete way to proceed. Maybe it is better to incorporate in a multivariable model what do you think could have a clinical effect on your outcome of interest. 

The reviewer highlights a very important point. We modified the univariate and multivariate analysis to incorporate version. And we modified the sentences associated with corrections.

Reviewer 2 Report (New Reviewer)

Comments and Suggestions for Authors

This retrospective study indicates that GNRI could stratify patients with high GNRI with low GNRI into high and low risk metastatic prostate cancer. This is not new (see reference 32 cited in this manuscript). What is new is the finding that hormone-sensitive high risk prostate cancer patients treated with abiraterone outperformed those treated with bicalutamide, especially when GNRI status was taken into account.

Author Response

Reviewer 2

This retrospective study indicates that GNRI could stratify patients with high GNRI with low GNRI into high and low risk metastatic prostate cancer. This is not new (see reference 32 cited in this manuscript). What is new is the finding that hormone-sensitive high risk prostate cancer patients treated with abiraterone outperformed those treated with bicalutamide, especially when GNRI status was taken into account.

In response to the reviewer’s recommendation, we highlighted the important point in this article as below:

“However, only one report exists of an association between high levels of GNRI and a better prognosis in mHSPC [32]. Our study is newly suggested the superiority of GNRI com-pared to NLR or CRP as biomarker in patients with LATITUDE high risk, and also sug-gested that the GNRI could be used to aid in patient selection in the upfront treatment of such patients. In addition, our data also suggest that all patients with LATITUDE high-risk and high GNRI levels should receive upfront AA with ADT instead of CAB as initial treatment. Furthermore, considering the poor prognosis, upfront AA with ADT should be strongly recommended for patients with low GNRI.” In Discussion.

Round 2

Reviewer 1 Report (Previous Reviewer 3)

Comments and Suggestions for Authors

None.

This manuscript is a resubmission of an earlier submission. The following is a list of the peer review reports and author responses from that submission.

Round 1

Reviewer 1 Report

Comments and Suggestions for Authors

Title: Geriatric Nutritional Risk Index as a Prognostic Indicator for

Japanese Patients with LATITUDE High-risk Metastatic Hormone- sensitive Prostate Cancer: A Multi-center Study

     Naiki et al. proposed GNRI as prognostic indicator for patients with mHSPC who received androgen deprivation therapy with abrirateron or bicalutamide by analyzing data on serum nutritional measurements and short-term treatment outcomes including CRPC diagnosis. However, albumin that is a part of GNRI formular is simple and more predictive than GNRI according to this study result. Therefore, it seems more appropriate to examine a prognostic value of albumin itself rather than GNRI. In addition, there is a lack of descriptions about data analysis methods such as ROC analysis. I have several comments as follows.

1.          Line 108-109:  it is written that if actual body weight was greater than ideal body weight the ratio of these two factors was set to be one.  Is this a part of the original definition of GNRI? If no, please justify it.

2.          Line 112: PFS was measured from the start of first treatment until the time of a CRPC diagnosis. Time to CRPC (TTCR) was determined to be the time interval between the introduction and failure of initial treatment.  PFS and TTCR seems to the same. What is a difference between them?   

3.          Line 117: Mann-Whitney U test could not be used to analyze a categorical parameter. Please clarify it.

4.          Lines 118: PFS is a time-to-event outcome. Did you use a time-dependent ROC?

5.          Line 147-150:  what outcome variable was used in ROC analysis? Also, there is no details about how to choose a new cutoff value for prognostic factors. For example, how did you choose the cutoff 109 for GNRI?

6.          Line 157-164: The observed significant differences in CRPC development and PFS seem to be just because the GNRI cutoff was selected based on CRPC diagnosis, which is the event for the PFS definition.

7.          Line 173-174: Again, the difference in PFS is simply because the albumin cutoff was selected by seeing PFS events (CRPC diagnosis).

8.          Line 180 and 181: I don’t think it is acceptable to take GNRI because Alb has a higher and better prognostic value of PFS than GNRI. This is a major and serious flaw in data analysis of this study.

9.          Line 190-205:  TTCR is defined as the time-to-CRPC which is also an treatment outcome and I think it is a strong precursor of overall survival.

10.      Line 194: Should OS be measured from the date of treatment with AA or CAB, not the date of diagnosis?

Reviewer 2 Report

Comments and Suggestions for Authors

Well described study.

Maybe a bit more relevant for Asian countries.

Follow up could be better but ok to describe the tendencies within mentioned limitations.

Reviewer 3 Report

Comments and Suggestions for Authors

In this study Authors aimed to test whether a Geriatric Nutritional Risk Index could represent a prognosticator of better survival in a Japanese cohort of patients with high-risk metastatic hormone-sensitive prostate cancer according to the LATITUDE criteria receiving either androgen deprivation therapy plus abiraterone acetate or androgen deprivation therapy plus bicalutamide. 

Despite the topic is interesting, I found difficulties in reading and interpreting the manuscript. This represents the main limitation of the paper. 

First, an extensive editing of the English language is strongly recommended. 

Second, a lot of analyses have been performed, a lot of results have been shown, and a lot of abbreviations have been used, thus causing the reader the loss of the focus on the purpose of the study. For example, if the study focuses on the importance of the above-mentioned index, all the results about the comparison between the two groups are not relevant. 

Third, how the Geriatric Nutritional Risk Index has been calculated? You showed the formula, but what are its scientific basis?

Authors should reorganize the manuscript structure in a more scientific way, presenting the methodology in a step wise fashion, and summarizing their results. 

Comments on the Quality of English Language

Extensive editing of English language is required.